# Thermal Aging Effects on the Mechanical Behavior of Glass-Fiber-Reinforced Polyphenylene Sulfide Composites

**DOI:** 10.3390/polym14071275

**Published:** 2022-03-22

**Authors:** Jiangang Deng, You Song, Zhuolin Xu, Yu Nie, Zhenbo Lan

**Affiliations:** 1Wuhan Nari Limited Liability Company of State Grid Electric Power Research Institute, Wuhan 430074, China; dengkelvin@163.com (J.D.); l17720559398@163.com (Y.S.); xuzhuolinwh@139.com (Z.X.); whnr18327057305@163.com (Y.N.); 2State Grid Electric Power Research Institute, Nanjing 210000, China

**Keywords:** GF/PPS, thermal aging, mechanical property, color difference

## Abstract

In this article, the thermal aging behavior of polyphenylene sulfide (PPS) composites, reinforced by 20% glass fibers (GFs), in thermal aging temperatures ranging from 85 to 145 °C was studied. Tensile and bending properties and color changes in the thermally aged samples were investigated. The results showed that thermal aging at this temperature range resulted in the degradation of mechanical properties. Both the tensile and flexural strength of the GF/PPS composites were significantly reduced after thermal aging at 145 °C. Decreased strength and brittle fracture were observed because thermal aging at high temperatures resulted in the deterioration of the interfaces between the GFs and PPS matrix. The degradation of the mechanical properties of the composite samples can be reflected by the color change, which means that the mechanical properties of the GF/PPS composite samples under thermal aging are predictable using color change analysis.

## 1. Introduction

Polyphenylene sulfide (PPS) is a high-performance engineering polymer with superior properties, which include very good physical, thermal, electrical, chemical, and mechanical properties. These physical and mechanical properties are directly related to its macromolecular structure, which consists of aromatic rings and divalent sulfur atoms [1,2,3]. It is used in many industries such as flame-retardant materials, mechanical and chemical engineering, electronics, and motor industries. In addition, PPS-based materials reinforced with glass fibers or carbon fibers have recently been frequently applied as high-performance materials, for example, as structural materials in aerospace and automobile industries and as electrodes and separators in other fields. In the last couple of years, many studies have focused on these aspects [4,5,6,7].

The mechanical and physical properties of pure PPS have been widely investigated; however, pure PPS has been rarely used as a final material because of its relatively low mechanical properties [8,9,10,11,12,13,14]. Therefore, researchers have developed a lot of PPS composites reinforced with fibers to improve the physical, thermal, and mechanical properties of PPS [15,16,17,18,19,20]. Among the group of reinforcing fibers, thermoplastic composites reinforced with continuous fibers are very interesting in terms of making large-size aircraft and other products. Among these reinforcements, glass fibers are the most popular, which were used for polymeric matrix composites in recent years. Glass fibers are normally used to strengthen the load capacity for the soft matrix because glass fibers have very high tensile strength along the length direction. The matrix generally provides good support for glass fibers and helps carry the load transfer. Glass-fiber-reinforced PPS (GF/PPS) composite material is one of the most successful PPS composite materials. Because of their superior comprehensive properties, GF/PPS composite materials are very promising composite materials for high-performance engineering applications and have been frequently used in fields such as aircraft and electronics.

However, the mechanical properties of the GF/PPS composites can be further improved. The mechanical properties of composite materials reinforced with fibers depend strongly on the characteristics of the fibers and matrix, the bond efficiency, and the interaction between them [21,22]. Several studies have been performed to investigate the physical properties and mechanical behavior of fiber-reinforced PPS [23,24,25,26,27,28]. It has been reported that thermal aging has a strong influence on both physical properties and mechanical properties. For instance, Scobbo et al. [29] investigated the effect of thermal aging on pure PPS material. The thermal aging treatments were carried out at a temperature range from 160 to 220 °C and held for 4 h. They reported that the modulus of the samples increased as the glass transition temperature increased. Pantelakis et al. investigated the thermal aging effect on tensile and shear properties of carbon-fiber-reinforced PPS composite samples. They reported that with increasing aging temperature and holding time, an obvious degradation of mechanical properties was observed [30]. Other researchers also investigated the mechanical degradation of PPS under thermal aging by using a variety of methods. Perng et al. [31] investigated the thermal decomposition of PPS and reported the relating decomposition mechanisms. Cao et al. [32] studied the expansion behavior and tensile properties of PPS composite materials after thermal cycling at high temperatures. They found that the crystallinity of PPS was increased in such treatments. Zuo et al. systematically investigated the influence of thermal aging on the physical and thermal properties and mechanical behaviors of GF/PPS composites [33,34]. They reported that strong degradation was observed in the composite samples after aging at 180 and 200 °C. The results implied that thermal aging treatments at such a temperature range have a strong influence on the crystallization behavior of PPS composites. Thermal aging at high temperatures generally leads to more reactions of crosslinking and a reduction in molecule mobility. The authors also studied the mechanical properties of thermally aged GF/PPS composites at a temperature range from 140 to 200 °C for a very long period of 5300 h. They reported that an increase in aging temperature resulted in high oxidation.

The thermal aging behavior of PPS–matrix composites is very important when they are used for structural applications at high temperatures. The high temperature may result in a sharp decrease in the mechanical properties of the composites. At high temperatures, the fiber-reinforced PPS–matrix composite may undergo significant chemical or structural changes, resulting in the serious degradation of mechanical properties. Very serious damaging phenomena, such as delamination or micro-cracking, can occur in these PPS composites when they are exposed to high temperatures and long periods.

Therefore, it is essential to understand the influence of high-temperature thermal aging on the mechanical properties of GF/PPS composites. In this study, the purpose was to detect the thermal aging effect on the mechanical properties of GF/PPS composite samples thermally aged at different temperatures. More importantly, the mechanical properties of the thermally aged GF/PPS composites were linked to their color changes after aging. Therefore, we could predict the mechanical properties of the thermally aged composite materials by analyzing their color change when comparing them with the unaged samples.

## 2. Materials and Methods

### 2.1. Materials

Glass-fiber-reinforced (20% weight fraction) PPS composites were used in the present study. The PPS powder was obtained from HongHe Company (Zigong, China). The glass fibers (GFs: FR5301B-2000) were obtained from Chongqing Polymer Composite International (Chongqing, China). Injected plate samples were manufactured with a barrel set temperature at 300 °C and a tool temperature at 100 °C. Dog-bone-shaped samples and plate samples were prepared for the tensile and bending test, respectively.

### 2.2. Thermal Aging Treatments

Thermal aging treatments were carried out in lab ovens (HZ-2004, Lyxyan, Dongguan, China) at temperatures ranging from 85 to 145 °C for a fixed holding time of 180 h. The temperature range selected in this study was chosen to investigate the thermal aging behavior below and above the glass transition temperature of the PPS matrix, which is approximately 90–100 °C [34,35]. The composite samples were put in a preheated furnace. The original sample without any thermal aging was also prepared for comparison. The composite samples were named GF/PPS-x samples, and x represented the holding temperature. For example, the GF/PPS-0 and GF/PPS-100 indicated the original sample and the sample after aging at 100 °C, respectively.

### 2.3. Mechanical Testing

The tensile properties of the samples were evaluated according to the ASTM D3039 standard. The tensile tests were performed at room temperature by using a tensile tester (MTS E45, MTS, Eden Prairie, MN, USA). A constant strain rate of 50 mm/min was used, which was measured using an extensometer. The tensile samples had a dimension of 2 mm × 10 mm × 150 mm, as shown in Figure 1a. The tensile strength, strain, and elastic modulus of the samples were evaluated five times. Average values and standard deviations of them were reported in this study.

The flexural properties of the samples were measured based on three-point bending mode as per ASTM D7264. The same machine (MTS E45, MTS, Eden Prairie, MN, USA) was used. A constant crosshead speed of 1 mm/min was used for bending tests of all the samples. The samples for flexural tests had a dimension of 80 mm × 10 mm × 4 mm and a support span length of 64 mm, as shown in Figure 1b. Similar to the tensile tests, the bending tests of the samples of each condition were repeated five times for average values and standard deviations. The flexural strength (*σ*), modulus (*E*), and strain to failure (*ε*) of the composites was calculated with Equations (1)–(3):(1)$$\sigma =3PL\;/\;2bh^{2}$$
(2)$$E=mL^{3}\;/\;4bh^{3}$$
(3)$$\varepsilon =6Dh\;/\;L^{2}$$
length, sample width, and sample thickness, *m* is the slope of the elastic portion of the load-strain curve, and *D* and *P* are the maximum deflection and the maximum load before failure, respectively.

### 2.4. Fracture Surface Analysis

For fracture surface analysis, the fractured tips were cut from the tensile-tested samples. The fracture surfaces were ultrasonically cleaned. The fracture surfaces of the samples were then coated with carbon and observed using field emission scanning electron microscopy (TESCAN MIRA III, Brno, Czech Republic). An acceleration voltage of 5 kV was used during the observation.

### 2.5. Color Characterization

The color changes of the GF/PPS composite samples after thermal aging were measured using a colorimeter (x-rite Color-Eye 7000A, Gretag Macbeth, Stevensville, MI, USA). The three values of *L*, *a*, and *b* were used to present the color change of the composites. *L* represents luminance, of which values range from 0 (dark) to 100 (bright). *a* and *b* indicate the changes from green to red and from blue to yellow, respectably. Their values range from −128 to 128. The color change, ΔE, can be calculated with Equation (4):(4)$$\Delta E={\left(\Delta L^{2}+\Delta a^{2}+\Delta b^{2}\right)}^{1/2}$$

Meanwhile, we also measured the grayscale and glossiness (*G*60) values for all the samples. Each sample was measured five times to obtain average values.

## 3. Results and Discussion

### 3.1. Tensile Properties

Tensile tests of the samples show the influence of thermal aging on the mechanical properties of the GF/PPS composites, as shown in Figure 2. Figure 2a shows the tensile stress–strain curves of the thermally aged samples. The original sample (GF/PPS-0) without thermal aging exhibited abnormal behavior, namely there was a very small region related to elastic deformation but a very large plastic region. The GF/PPS-85 sample exhibited similar behavior to that of the GF/PPS-0 sample but gave slightly reduced tensile strength (Figure 2a). With increasing thermal aging temperature, the GF/PPS-100 to −145 samples exhibited normal deformation behavior, which is typical of an elastic region followed by plastic deformation until fracture. Figure 2b–d show the tensile properties of the samples as a function of aging temperature. As shown in Figure 2b, the sample strengths decreased at a higher aging temperature. The tensile strain exhibited almost the same tendency, except for the GF/PPS-85 sample, which was thermally aged at a temperature below the glass transition temperature. As shown in Figure 2c, the samples treated above the glass transition temperature exhibited a large decrease in tensile strain. However, it was shown that the elastic modulus did not change significantly with increasing aging temperature, as shown in Figure 2d.

Summarily, thermal aging led to the significant degradation of the mechanical properties of all the aged samples, especially for the large loss of ductility. The mechanical properties of GF/PPS composites generally depend on the interaction between PPS and GFs, as well as the crystallinity of PPS. The interfacial adhesion between GFs and PPS is gradually weakened with increasing aging temperature. In addition, the PPS oxidation may also affect the crystallinity and the change in interaction between PPS resin and GFs during high-temperature aging. This is because the linear or branched molecular structures of thermoplastics become more brittle during thermal aging [36]. Researchers attributed this to the cross-linking and crystallinity, which depended on the temperature and the exposure time, as reported by Zhang et al. [37] and Lee et al. [38]. Molecular chain scission, post-crosslinking, and crystallinity increment also occur in the GF/PPS composite samples during thermal aging [39].

### 3.2. Failure Analysis

The fractured surfaces observed using SEM and the micrographs are presented in Figure 3 and Figure 4. Figure 3 shows the overall fracture surfaces of the composites. It is observed that the fracture surface exhibited very rough features. There were no obvious features regarding the initiation or propagation of cracks, indicating the high toughness of the GF/PPS composite samples. In addition, there was no significant difference between the composite sample aging at different temperatures. As a result, high-magnification images of the fracture surfaces were taken to study the failure behavior of the composite samples after aging.

Figure 4 shows high-magnification images of the composite samples after a tensile fracture. The fracture surface of the unaged sample in Figure 4a shows a high quality of adhesion between GFs and PPS. PPS resin thin layers were observed on the GF surfaces (indicated by arrows). This indicated good interfacial adhesion of the PPS and GFs in the unaged sample. For the GF/PPS composites after aging, however, very smooth surfaces of GFs were observed. In the GF/PPS composites after aging at high temperatures, obvious separations between GFs and PPS were observed. Therefore, the tensile strength degradation of the thermally aged composites is associated with the weakened interfacial bonding between the GFs and PPS because of PPS degradation [40,41]. The large and prevalent gaps between GFs and PPS resulted in further interface bonding degradation. There was no significant damage to GFs; however, the smooth cross-sectional surfaces of the fibers indicated a brittle failure. This is because of the ductility loss of fibers. Comparatively, in the case of the aged sample at 145 °C, Figure 4f, smooth surfaces of fibers were frequently observed. This indicates serious degradation of the PPS matrix. Thermal aging damaged the cohesion between the matrix and fibers and resulted in separation. As a result, the sample aged at 145 °C exhibited the lowest mechanical properties.

### 3.3. Flexural Properties

The flexural properties of the samples are shown in Figure 5. As shown in Figure 5a, the GF/PPS-0 composite sample exhibited a sharp stress increase in the beginning and showed the highest flexural strength compared to the aged samples. By contrast, the GF/PPS composite samples after thermal aging showed a slow decrease in stress. Figure 5b,c show the effect of aging temperature on flexural strength and the modulus of the aged composites. The results revealed that the flexural strength decreased from 200 to 100 MPa, whereas the flexural modulus decreased from 34 GPa to 21.4 GPa with increasing aging temperature up to 145 °C. The GF/PPS-145 sample exhibited the lowest strength and strain among all the tested samples. This is because of the PPS matrix degradation under thermal aging. These findings are in good agreement with the tensile result, as shown in Figure 2.

### 3.4. Color Difference

Table 1 shows the color change of the samples before and after aging. There was no significant grayscale change observed in the aged samples as compared to the original sample. This may be because the holding temperatures used in this study were selected around their glass transition temperature, approximately 90–100 °C [34,35]. Therefore, the GF/PPS composites were not easily oxidized during the aging process. However, the *ΔE* and *G*60 of the GF/PPS composites did change significantly. For example, with increasing aging temperature from 85 to 145 °C, the ΔE value decreased from 4.11 to 1.74. In addition, the gloss (G*60*) of the composite samples also sharply decreased from 56.6 to 8.4 after aging at 85 °C and gradually decreased to approximately 4.4 at higher aging temperatures.

Overall, the results from the measurement of Δ*E* and *G60* values agreed well with the change in mechanical properties of the sample after aging. The *ΔE* and *G60* values and mechanical properties of the samples are plotted in Figure 6. The decrease in *ΔE* and *G60* values gradually decreased accompanied by the mechanical properties. A good correlation between the color difference (Δ*E* and *G60* values) and the mechanical properties (tensile strength and flexural strength) of the GF/PPS composite samples was observed. This implies that using color change analysis enables the prediction of both the tensile and flexural strengths of the thermally aged GF/PPS composite materials. In practice, it is a very useful method to indirectly predict the degradation of composite materials and their service life.

## 4. Conclusions

In this study, PPS composites reinforced with 20% glass fiber were thermally aged at a temperature ranging from 85 to 145 °C to investigate the thermal aging performance in high temperatures. The tensile and flexural properties and color change behavior of the thermally aged GF/PPS composites were studied. The results showed that both the tensile and flexural strength of the composite samples strongly decreased with increasing aging temperature. The tensile strength and flexural strengths of the GF/PPS composites after thermal aging decreased from 130 MPa to 70 MPa and 200 MPa to 99 MPa, respectively. The high-temperature aging also resulted in the loss of tensile ductility from 7% to 3%. The elastic modulus of the samples was maintained in the range of 5−6 GPa, which is almost independent of the aging temperature. The fracture surface analysis of thermally aged composites revealed obvious brittle fractures because of the interfacial bonding degradations between GFs and PPS matrix. The decrease in tensile strength and flexural strength with increasing aging temperature was well reflected by the color change. The tensile and flexural strength exhibited the same tendency as the values of Δ*E* and *G60* with increasing temperature. Based on the correlation between the color difference and mechanical properties, the mechanical properties of thermally aged samples are predictable when color difference analysis is used. This work provided a simple method to predict their mechanical properties during actual applications.

## Figures and Tables

**Figure 1 polymers-14-01275-f001:**
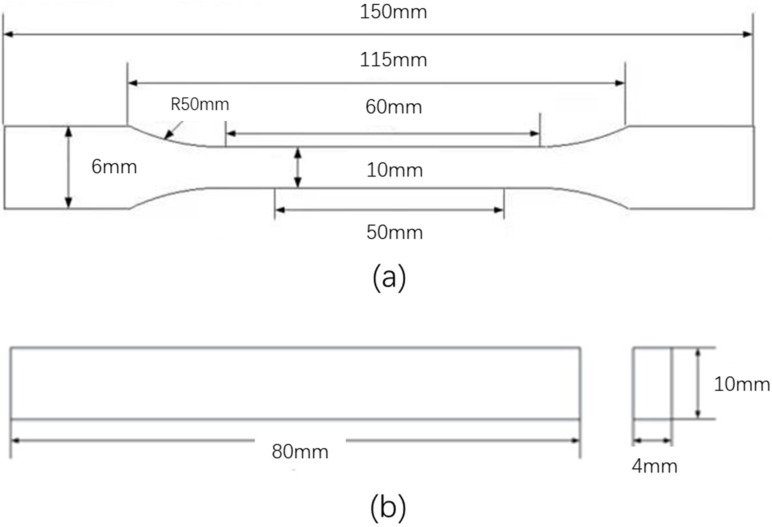
Dimensions of the specimens for (**a**) tensile test and (**b**) bending test.

**Figure 2 polymers-14-01275-f002:**
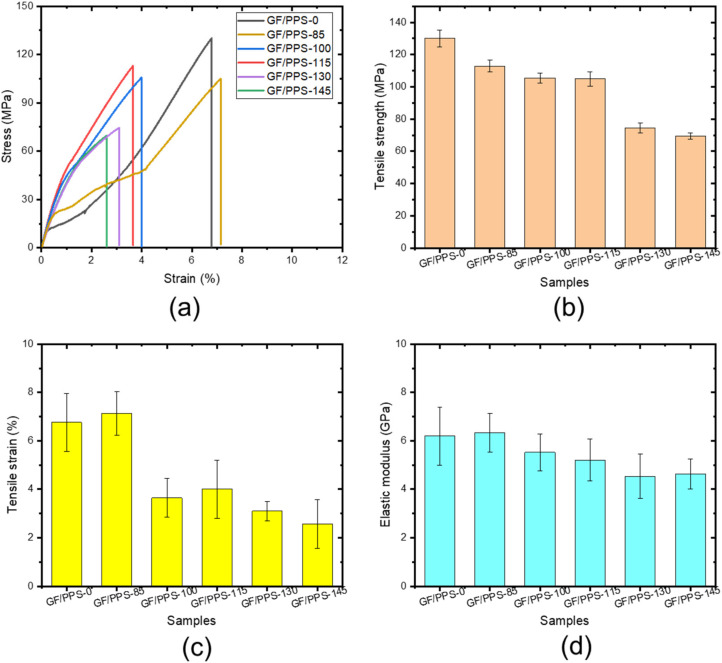
Tensile stress–strain curves (**a**), tensile strength (**b**), tensile strain (**c**), elastic modulus (**d**) of the thermally aged samples.

**Figure 3 polymers-14-01275-f003:**
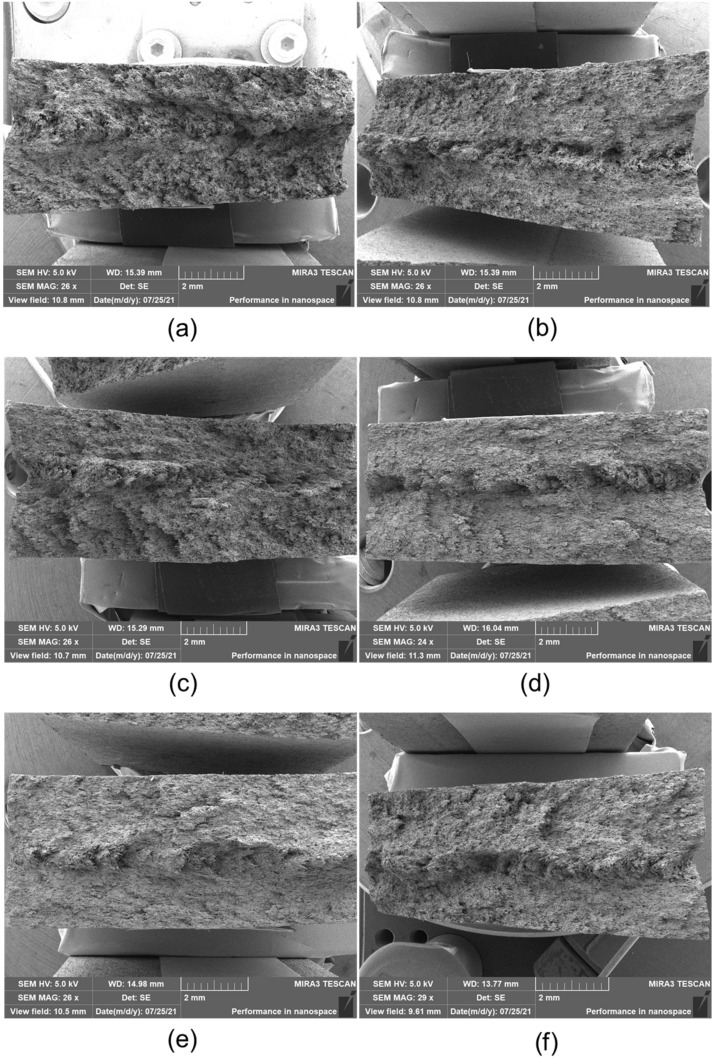
Fracture surface observations of the composite samples: (**a**) GF/PPS-0, (**b**) GF/PPS-85, (**c**) GF/PPS-100, (**d**) GF/PPS-115, (**e**) GF/PPS-130, (**f**) GF/PPS-145.

**Figure 4 polymers-14-01275-f004:**
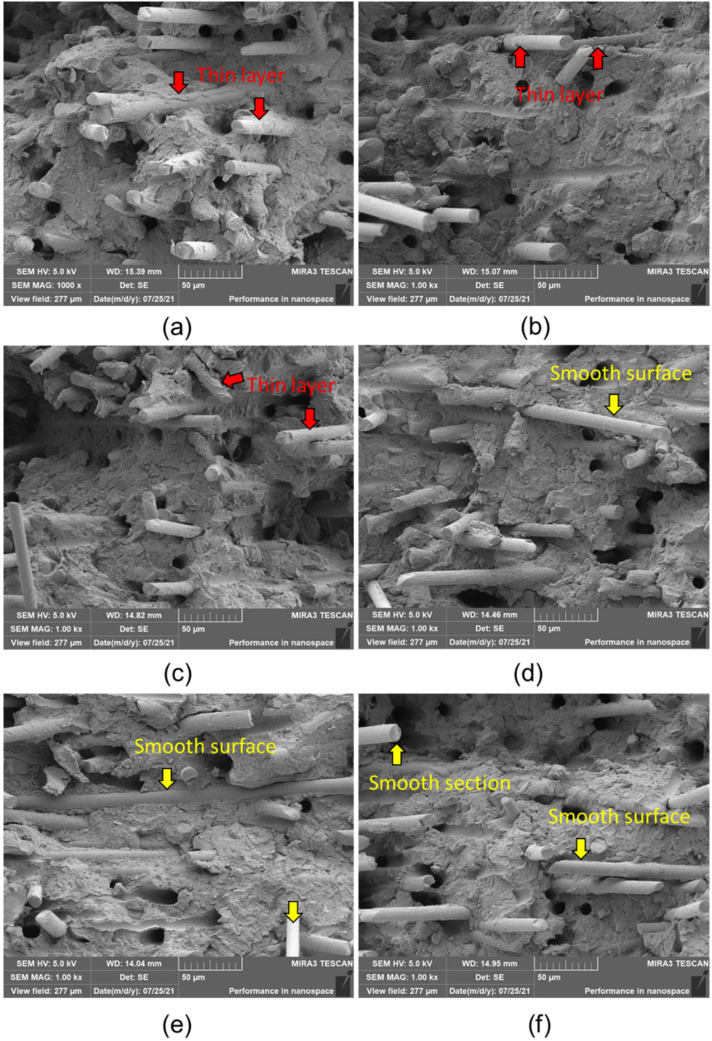
High-magnification images of the fracture surfaces: (**a**) GF/PPS-0, (**b**) GF/PPS-85, (**c**) GF/PPS-100, (**d**) GF/PPS-115, (**e**) GF/PPS-130, (**f**) GF/PPS-145.

**Figure 5 polymers-14-01275-f005:**
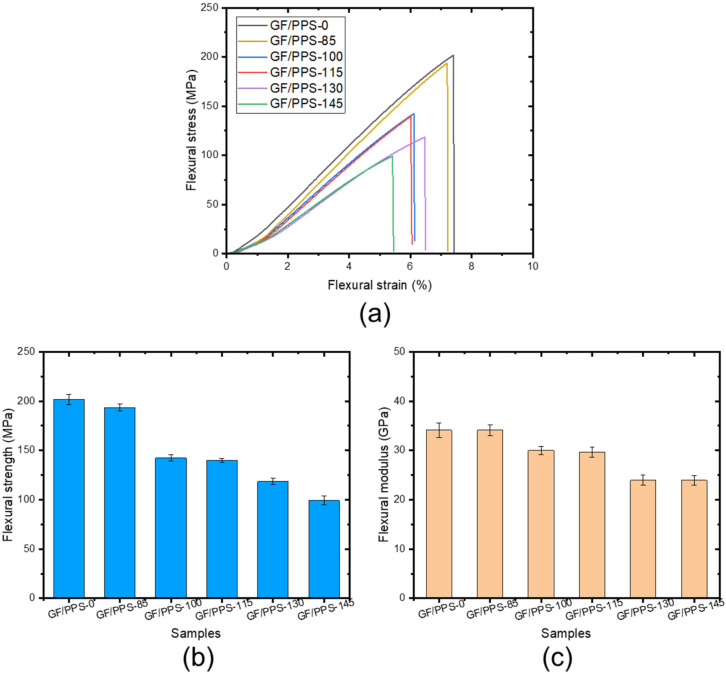
Flexural stress–strain curves of the samples (**a**), flexural strength (**b**), and flexural modulus (**c**) of the aged samples as a function of temperature.

**Figure 6 polymers-14-01275-f006:**
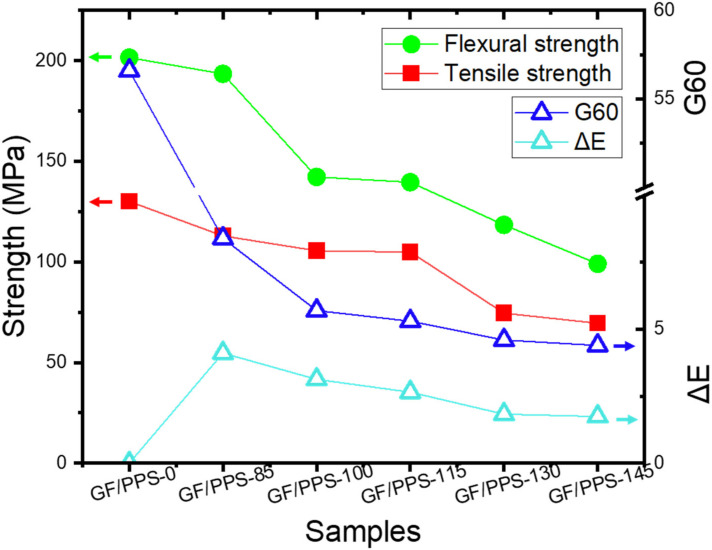
Correlation between mechanical strength and color difference of the GF/PPS samples.

**Table 1 polymers-14-01275-t001:** Color difference of the GF/PPS composites after thermal aging.

Samples	*L*	*a*	*b*	Δ*E*	Gray	*G*60
GF/PPS-0	81.56	0.48	13.12	0	202 ± 2	56.6 ± 3.3
GF/PPS-85	82.81	0.48	9.21	4.11 ± 0.23	206 ± 4	8.4 ± 1.3
GF/PPS-100	82.92	0.39	10.30	3.13 ± 0.28	206 ± 5	5.7 ± 1.2
GF/PPS-115	82.68	0.43	10.72	2.65 ± 0.31	205 ± 3	5.3 ± 1.5
GF/PPS-130	82.44	0.84	11.56	1.83 ± 0.22	204 ± 3	4.6 ± 1.2
GF/PPS-145	83.19	0.40	12.54	1.74 ± 0.27	207 ± 2	4.4 ± 1.0

## Data Availability

All data used during the study appear in the submitted article.

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
