# Peer review of "Thermal Aging Effects on the Mechanical Behavior of Glass-Fiber-Reinforced Polyphenylene Sulfide Composites"

_polymers, 2022, doi:10.3390/polym14071275_

Round 1

Reviewer 1 Report

The submitted manuscript is the resubmission of the previously reviewed manuscript polymers-1565642. The authors provided detailed response to each comment and made significant changes in the manuscript, which sufficiently improved it. I have no other comments and recommend this manuscript for a publication in its present form.

Author Response

We thank the reviewer for these positive comments.

Reviewer 2 Report

The manuscript, entitled “Thermal Aging Effects on the Mechanical Behavior of Glass Fiber-Reinforced Polyphenylene Sulfide Composites”, by Jiangang Deng, You Song, Zhuolin Xu, Yu Nie, and Zhenbo Lan, presented a study on thermal aged fiber reinforced PPS composites, combining both mechanical testing and color detection. The authors observed the weakening of mechanical performances, microstructural evolution, and color change. However, the authors still need to revise it to make the manuscript more suitable for the publication in Polymers (MDPI) journal.

  1. Introduction

Compared to the previous version, this new submission presents a much stronger motivation. However, the authors still need to clarify the motivation.

  • From Line 50-76, the authors reviewed many thermal aging study on PSS/GF composites, but in Line 81-83, the authors commented that “very few studies are discussing the effect of high temperature thermal aging on the physical, thermal and mechanical properties of GF/PPS composites”. This statement is confusing considering the number of previous papers published in this field. The authors need to make more cohesive, consistent statements.
  • The authors also need to edit Line 52-55, because the authors stated that “thermal aging has a strong influence”, but afterward, the example is about the influence of glass fiber loading on modulus. These few sentences are not consistent. Please use the example to clarify thermal aging effect.
  1. Result interpretation
  • The interpretation of color indicator is not clear to me. The G60 does seem to be an indicator, but ΔE does not seem to be a good one. In Line 235, the authors mentioned that ΔE increased from 4.11 to 1.74. Is this a typo?
  • Would be nice to include toughness data as well (basically integrating stress-strain curve from tensile loading)
  1. Typos/Corrections

There are many many minor corrections the authors need to do. Please proof-read it!

  • The mixed usage of past and present senses is a huge issue. For example, Line 37, “the most popular one that was used for polymer matrix” should be “the most popular one that is used for polymer matrix”. Other verbs are in present sense in this paragraph. The authors have made the same mistake in many other places, including Introduction and the rest of the manuscript.
  • Plural and single mixed usage. For example, Line 47, “depends” should be corrected as “depend”. The authors made many similar mistakes
  • Other issues, for example, Line 113, “were evaluated with five times” should be “were evaluated for five times”. Again, please proof-read.

Round 2

Reviewer 2 Report

I appreciate the authors' revision. I think it is suitable to be published in Polymers. I don't have additional comments regarding this manuscript.

This manuscript is a resubmission of an earlier submission. The following is a list of the peer review reports and author responses from that submission.

Round 1

Reviewer 1 Report

In the present study, the authors have examined the effect of thermal aging of glass-fiber (GF) reinforced polyphenylene sulfide (PPS) composites with respect to tensile strength, flexural strength, and color change behavior. The results indicate decrease in both tensile strength and flexural strength of the GF/PPS composites with a correlated change in color. The manuscript is well written and nicely organized. However, I would recommend the authors to address the following comments before the publication of the manuscript in the journal of Polymers.

  1. On page 2 and in lines 58-87, the authors discuss several past studies that have examined the effect of thermal aging on GF/PPS composites, so it is not clear to me how the present study is different from earlier literature. I would highly recommend the authors to clarify this in the introduction. This would help the readers to appreciate the significance of the present work in context of earlier studies.
  2. The authors use 85, 100, 115, 130 and 145 C temperatures and a duration of 180 h for thermal aging . It would be helpful if the authors clarify as to why they chose these aging temperatures and time for their study.
  3. For both tensile and flexural mechanical measurement results presented in Figures 2 and 5, the authors make comparisons across non-aged and aged samples. However, these comparisons are not justified without performing a rigorous statistical analysis. For example, it is hard to say whether the tensile strain has reduced after aging the samples at 85C in Figure 2c considering the error bars associated with the measurement. Therefore, the authors should perform statistical analysis and report p-values to make meaningful comparisons across fresh and aged samples.
  4. The authors should clarify in the materials and methods section as to what the error bars represent. Are these standard deviations or standard errors?
  5. It would be helpful if the authors also provide some SEM images of simply glass fibers in the Supporting Information to show their smooth surface (as a control).

Reviewer 2 Report

The authors of the submitted manuscript presented the results on investigation of an influence thermal aging on mechanical properties of GF/PPS composites. The Introduction presents a well prepared overview of the problem of thermal aging and its characterization, however, some minor recommendations were provided below. In section 2, the authors described the tested specimens as well as their characterization procedures. Further, in section 3, the results were presented.

The manuscript presents typical studies of evaluation of aging of polymers and composites, and the originality need to be emphasized.

1) Lines 90-91: probably the last word in this sentence should be “properties” or similar, instead of “temperature”.

2) In the Introduction the authors can put more attention on DMA and DSC tests that are able to define well the mechano-chemical changes in polymers and polymer matrix composites, including thermal aging. Moreover, it would be beneficial to define open problems resulting from literature review and link them with the defined research problem in the manuscript. The emphasis of originality of this study is needed.

3) Please justify the selection of temperature and duration of ageing described in section 2.2. Is it connected with a specific process? In particular, it is of high interest what is the glass-transition temperature of the investigated polymer. It is necessary to link the observed results described in section 3 with this temperature.

4) It would be beneficial to compare and support the explanations of the observed phenomena presented in section 3 with independent studies from literature.

5) In Figure 6, it is better to add legend rather than captions on particular curves.

6) It is recommended to enrich the Conclusions with quantitative results obtained within the study.

Reviewer 3 Report

Regarding your submission of the manuscript “Thermal Aging Effects on the Mechanical Behavior of Glass Fiber-Reinforced Polyphenylene Sulfite Composites” by Jiangang Deng, You Song, Zhuolin Xu, Yu Nie, and Zhenbo Lan, I have some comments listed below. Overall, this thermal aging study on glass fiber filled PPS was complete and cohesive. The effects of thermal aging on mechanical performances, microstructures, and colors were clearly conveyed. I believe this could be meaningful for polymer composite technical community.

However, there are some revisions that could be done to improve the quality of this manuscript:

  1. Writing and grammars.

There are some grammatical issues that could be worked on. Overall, these issues don’t affect my understanding too much, but it is still much nicer to do a thorough grammatical checking. A few examples are here:

  • Line 26-27, “PPS-based materials… … used as a high-performance material”. This sentence could be corrected as “PPS-based materials… … used as high-performance materials”.
  • Line 29-30, “the last couple of years… …are focusing on these aspects.” Correcting it to “were focused on”, or “have been focused on” will be better.
  • Line 67, “220 C and hold for 4 h”, should be corrected as “220C and held for 4 h”.
  • Line 161, “At meanwhile”, it is more common to say “Meanwhile”, or “In the meanwhile”.

There are many examples like these. Hope these issues could be properly addressed. 

  1. Clarifications on experimental details
  • You have mentioned that many researchers conducted thermal aging experiments at 180 °C, 200 °C, 140 °C, etc (Line 80-81, 86), any reasons for choosing lower temperatures for your study compared with other study? (85-145 °C)
  • The materials you used contained 20% glass fibers, by weight? By volume? It was not clear to me.
  1. Result interpretation
  • You might have cited literature in introduction session, but there are some analysis and statements in the “Results and discussion” session that need further literature support. For example, line 182-183, any reference to support your statement on chain scission and crystallization? Adding a few citations from your previous introduction can make this statement stronger.
  • Can you elaborate more on the difference between grayscale and G60 change, and cite literature properly to prove your statement on oxidation? (Line 251-266). This paragraph was not clear to me if oxidation is easy or not.
  • In addition to previous point, PPS glass transition temperature is about 90 °C, why did you comment that “the holding temperatures used were quite lower than their glass transition temperatures” (line 256). Aren’t 85-145 °C much higher than 90 °C? Is that a typo? Please include Tg of the matrix in the manuscript.
  1. Modifications on plots/tables
  • You described the thin layer of coating on fibers (line 214) and smooth cross-sectional surface (line 225) in Figure 4. Can you add arrows or texts on the images to point those out? That would be more indicative.

Author Response

We thank the editor and reviewer for their consideration of our work and for the time and energy they have expended to provide feedback on our manuscript. We have revised the manuscript in response to reviewer’s valuable comments. The details of our responses are described below.

<Reviewer 1>

In the present study, the authors have examined the effect of thermal aging of glass-fiber (GF) reinforced polyphenylene sulfide (PPS) composites with respect to tensile strength, flexural strength, and color change behavior. The results indicate decrease in both tensile strength and flexural strength of the GF/PPS composites with a correlated change in color. The manuscript is well written and nicely organized. However, I would recommend the authors to address the following comments before the publication of the manuscript in the journal of Polymers.

Response >:

We thank the reviewer for these positive comments.

 1. On page 2 and in lines 58-87, the authors discuss several past studies that have examined the effect of thermal aging on GF/PPS composites, so it is not clear to me how the present study is different from earlier literature. I would highly recommend the authors to clarify this in the introduction. This would help the readers to appreciate the significance of the present work in context of earlier studies.

Response>:

We thank the reviewer for the very helpful comments. The novelty of the present study is that we try to correlate mechanical properties and color change of the GF/PPS composites after thermal aging. We hope that we can predict the mechanical properties of the composite materials after thermal aging based on the color change analysis. We believe that this is very suitable for actual application of the composites.

Revised content:<page 2>

More importantly, the mechanical properties of the thermally aged GF/PPS composites were linked to their color changes after aging. Therefore, we can predict the mechanical properties of the thermally aged composite materials through analyzing their color change when comparing them with the unaged samples.

2. The authors use 85, 100, 115, 130 and 145 C temperatures and a duration of 180 h for thermal aging. It would be helpful if the authors clarify as to why they chose these aging temperatures and time for their study.

Response>:

This comment is very important. The selected temperature in the present study is based on the glass transition temperature, which is approximately 90-100 ⁰C. As a result, the temperatures ranging from 85 to 145 ⁰C were selected to study the thermal aging behavior below and above this temperature.

Revised content:<page 3>

The temperature range selected in this study was to investigate the thermal aging behavior below and above the glass transition temperature of PPS matrix, which is approximately 90-100 °C [34,35].

3. For both tensile and flexural mechanical measurement results presented in Figures 2 and 5, the authors make comparisons across non-aged and aged samples. However, these comparisons are not justified without performing a rigorous statistical analysis. For example, it is hard to say whether the tensile strain has reduced after aging the samples at 85C in Figure 2c considering the error bars associated with the measurement. Therefore, the authors should perform statistical analysis and report p-values to make meaningful comparisons across fresh and aged samples.

Response>:

This is a helpful comment. We apologize that we were unable to perform a rigorous statistical analysis due to equipment problems; however, the reported strength and strain values were averages of five samples. GF/PPS-85 sample was thermally aged at a temperature below the glass transition temperature, therefore the tensile strain did not change significantly. the samples treated at above the glass transition temperature exhibited large decrease in tensile strain as shown in Figure 2c.

Revised content:<page 5>

The tensile strain exhibited almost the same tendency, except for the GF/PPS-85 sample which was thermally aged at a temperature below the glass transition temperature. As shown in Figure 2c, the samples treated at above the glass transition temperature exhibited large decrease in tensile strain.

4. The authors should clarify in the materials and methods section as to what the error bars represent. Are these standard deviations or standard errors?

Response>:

This is a good suggestion. The error bars represent standard deviations. We changed the text accordingly.

Revised content:<page 3>

The tensile strength, strain, and elastic modulus of the samples were evaluated with five times. Average values, and standard deviations of them were reported in this study.

Similar to the tensile tests, the bending tests of the samples of each condition were re-peated five times for average values and standard deviations.

5. It would be helpful if the authors also provide some SEM images of simply glass fibers in the Supporting Information to show their smooth surface (as a control).

Response>:

This is a good suggestion. We apologize that currently we could not find the original glass fiber because all the glass fibers we had were used up during the composite manufacturing.

<Reviewer 2>

The authors of the submitted manuscript presented the results on investigation of an influence thermal aging on mechanical properties of GF/PPS composites. The Introduction presents a well prepared overview of the problem of thermal aging and its characterization, however, some minor recommendations were provided below. In section 2, the authors described the tested specimens as well as their characterization procedures. Further, in section 3, the results were presented.

Response >:

We thank the reviewer for these positive comments.

The manuscript presents typical studies of evaluation of aging of polymers and composites, and the originality need to be emphasized.

1) Lines 90-91: probably the last word in this sentence should be “properties” or similar, instead of “temperature”.

Response>:

Thank you for your comment. The text has been replaced.

Revised content:<page 2>

The thermal aging behavior of PPS–matrix composites is very important when they are used as structural applications at high temperatures. The high temperature may result in a sharp decrease in the mechanical properties of the composites. At high temperatures, the fiber-reinforced PPS–matrix composite may undergo significant chemical or structural changes, resulting in serious degradation of mechanical proper-ties. Very serious damaging phenomena, such as delamination or micro-cracking can occur in these PPS composites when they are exposed to high temperatures and long periods.

2) In the Introduction the authors can put more attention on DMA and DSC tests that are able to define well the mechano-chemical changes in polymers and polymer matrix composites, including thermal aging. Moreover, it would be beneficial to define open problems resulting from literature review and link them with the defined research problem in the manuscript. The emphasis of originality of this study is needed.

Response>:

We thank the reviewer for the very helpful comments. The novelty of the present study is that we try to correlate mechanical properties and color change of the GF/PPS composites after thermal aging. We hope that we can predict the mechanical properties of the composite materials after thermal aging based on the color change analysis. We believe that this is very suitable for actual application of the composites.

Revised content:<page 2>

More importantly, the mechanical properties of the thermally aged GF/PPS composites were linked to their color changes after aging. Therefore, we can predict the mechanical properties of the thermally aged composite materials through analyzing their color change when comparing them with the unaged samples.

3) Please justify the selection of temperature and duration of ageing described in section 2.2. Is it connected with a specific process? In particular, it is of high interest what is the glass-transition temperature of the investigated polymer. It is necessary to link the observed results described in section 3 with this temperature.

Response>:

This comment is very important. The selected temperature in the present study is based on the glass transition temperature, which is approximately 90-100 ⁰C. As a result, the temperatures ranging from 85 to 145 ⁰C were selected to study the thermal aging behavior below and above this temperature.

Revised content:<page 2>

The temperature range selected in this study was to investigate the thermal aging behavior below and above the glass transition temperature of PPS matrix, which is approximately 90-100 °C [34,35].

4) It would be beneficial to compare and support the explanations of the observed phenomena presented in section 3 with independent studies from literature.

Response>:

This comment is very important. We have modified accordingly.

Revised content:<page 5>

Summarily, thermal aging led to significant degradation of the mechanical properties of all the aged samples, especially for the large loss of ductility. The mechanical properties of GF/PPS composites generally depend on the interaction between PPS and GFs, as well as the crystallinity of PPS. The interfacial adhesion between GFs and PPS is gradually weakened with increasing aging temperature. In addition, the PPS oxidation may also affect the crystallinity and the change of interaction between PPS resin and GFs during high-temperature aging. This is because of that the linear or branched molecular structures of thermoplastics become more brittle during thermal aging [36]. Researchers attributed this to the cross-linking and crystallinity, which depended on the temperature and the exposure time, as reported by Zhang et al [37] and Lee et al [38]. Molecular chain scission, post-crosslinking, and crystallinity increment also occur in the GF/PPS composite samples during thermal aging [39].

5) In Figure 6, it is better to add legend rather than captions on particular curves.

Response>:

Thank you for the suggestion. The Figure 6 has been replaced.

6) It is recommended to enrich the Conclusions with quantitative results obtained within the study.

Response>:

Thank you for the suggestion. The conclusion section has been modified as per your suggestion.

Revised content:<page 10-11>

In this study, PPS composites reinforced with 20% glass fiber, were thermally aged at a temperature ranging from 85 to 145 °C to investigate the thermal aging performance in high temperatures. The tensile and flexural properties and, color change behavior of the thermally-aged GF/PPS composites were studied. The results showed that both the tensile and flexural strength of the composite samples strongly decreased with increasing aging temperature. The tensile strength and flexural strengths of the GF/PPS composites after thermal aging decreased from 130 MPa to 70 MPa, 200 MPa to 99 MPa, respectively. The high temperature aging also resulted in the loss of tensile ductility from 7% to 3%. The elastic modulus of the samples maintained in the range of 5−6 GPa, which are almost independent on the aging temperature. The fracture surface analysis of thermally aged composites revealed obvious brittle fractures, because of the interfacial bonding degradations between GFs and PPS matrix. The decrease of tensile strength and flexural strength with increasing aging temperature is well reflected by the color change. The tensile and flexural strength exhibited the same tendency as the values of ΔE and G60 with increasing temperature. Based on the correlation between the color difference and mechanical properties, the mechanical properties of thermally aged samples are predictable to use color difference analysis. This work provided a simple method to predict their mechanical properties during actual applications.

<Reviewer 3>

Regarding your submission of the manuscript “Thermal Aging Effects on the Mechanical Behavior of Glass Fiber-Reinforced Polyphenylene Sulfite Composites” by Jiangang Deng, You Song, Zhuolin Xu, Yu Nie, and Zhenbo Lan, I have some comments listed below. Overall, this thermal aging study on glass fiber filled PPS was complete and cohesive. The effects of thermal aging on mechanical performances, microstructures, and colors were clearly conveyed. I believe this could be meaningful for polymer composite technical community.

However, there are some revisions that could be done to improve the quality of this manuscript:

Response >:

We thank the reviewer for these positive comments.

  1. Writing and grammars.

There are some grammatical issues that could be worked on. Overall, these issues don’t affect my understanding too much, but it is still much nicer to do a thorough grammatical checking. A few examples are here:

  • Line 26-27, “PPS-based materials… … used as a high-performance material”. This sentence could be corrected as “PPS-based materials… … used as high-performance materials”.
  • Line 29-30, “the last couple of years… …are focusing on these aspects.” Correcting it to “were focused on”, or “have been focused on” will be better.
  • Line 67, “220 C and hold for 4 h”, should be corrected as “220C and held for 4 h”.
  • Line 161, “At meanwhile”, it is more common to say “Meanwhile”, or “In the meanwhile”.

There are many examples like these. Hope these issues could be properly addressed.

 Response >:

We thank the reviewer for these comments. The texts have been corrected and highlighted in revised manuscript.

  1. Clarifications on experimental details
  • You have mentioned that many researchers conducted thermal aging experiments at 180 °C, 200 °C, 140 °C, etc (Line 80-81, 86), any reasons for choosing lower temperatures for your study compared with other study? (85-145 °C)

Response>:

This comment is very important. The selected temperature in the present study is based on the glass transition temperature, which is approximately 90-100 ⁰C. As a result, the temperatures ranging from 85 to 145 ⁰C were selected to study the thermal aging behavior below and above this temperature.

Revised content:<page 3>

The temperature range selected in this study was to investigate the thermal aging behavior below and above the glass transition temperature of PPS matrix, which is approximately 90-100 °C [34,35].

  • The materials you used contained 20% glass fibers, by weight? By volume? It was not clear to me.

Response>:

Thank you for the comment. Weigh prevent of 20% glass fibers were used. The corresponding text has been changed.

Revised content:<page 3>

Glass fibers (20% weight fraction) reinforced PPS composites were used in the present study.

  1. Result interpretation
  • You might have cited literature in introduction session, but there are some analysis and statements in the “Results and discussion” session that need further literature support. For example, line 182-183, any reference to support your statement on chain scission and crystallization? Adding a few citations from your previous introduction can make this statement stronger.

Response>:

Thank you for the comment. We have modified accordingly.

Revised content:<page 5>

Summarily, thermal aging led to significant degradation of the mechanical properties of all the aged samples, especially for the large loss of ductility. The mechanical properties of GF/PPS composites generally depend on the interaction between PPS and GFs, as well as the crystallinity of PPS. The interfacial adhesion between GFs and PPS is gradually weakened with increasing aging temperature. In addition, the PPS oxidation may also affect the crystallinity and the change of interaction between PPS resin and GFs during high-temperature aging. This is because of that the linear or branched molecular structures of thermoplastics become more brittle during thermal aging [36]. Researchers attributed this to the cross-linking and crystallinity, which depended on the temperature and the exposure time, as reported by Zhang et al [37] and Lee et al [38]. Molecular chain scission, post-crosslinking, and crystallinity increment also occur in the GF/PPS composite samples during thermal aging [39].

  • Can you elaborate more on the difference between grayscale and G60 change, and cite literature properly to prove your statement on oxidation? (Line 251-266). This paragraph was not clear to me if oxidation is easy or not.

Response>:

Thank you for the comment. We have modified this paragraph to make our points clear. We apologize that we could not find any literature regarding the correlation between mechanical properties and color difference of PPS based materials.

Revised content:<page 9-10>

Table 1 shows the color change of the samples before and after aging. There was no significant grayscale change observed in the aged samples as compared to the original sample. This may because that the holding temperatures used in this study were selected around their glass transition temperature, approximately 90-100 °C [34,35]. Therefore, the GF/PPS composites are not easily oxidized during the aging process. However, the ΔE and G60 of the GF/PPS composites did change significantly. For ex-ample, with increasing aging temperature, the ΔE value increased from 4.11 to 1.74. In addition, the gloss (G60) of the composite samples also sharply decreased from 56.6 to 8.4 after aging at 85 °C, and gradually decreases to approximately 4.4 at higher aging temperatures.

Overall, the results from the measurement of ΔE and G60 values agrees well with the change of mechanical properties of the sample after aging. The ΔE and G60 values, and mechanical properties of the samples were plotted in Figure 6. The decrease of ΔE and G60 values gradually decreased accompanied by the mechanical properties. A good correlation between the color difference (ΔE and G60 values) and the mechanical properties (tensile strength and flexural strength) of the GF/PPS composite samples was observed. This implies that using color change analysis enable to predict both the tensile and flexural strengths of the thermally aged GF/PPS composite materials. In practice, it is a very useful method to indirectly predict the degradation of composite materials and their service life.

  • In addition to previous point, PPS glass transition temperature is about 90 °C, why did you comment that “the holding temperatures used were quite lower than their glass transition temperatures” (line 256). Aren’t 85-145 °C much higher than 90 °C? Is that a typo? Please include Tg of the matrix in the manuscript.

Response>:

Thank you for the comment. We apologize for this typo.

Revised content:<page 9>

This may because that the holding temperatures used in this study were selected around their glass transition temperature, approximately 90-100 °C [34,35].

  1. Modifications on plots/tables
  • You described the thin layer of coating on fibers (line 214) and smooth cross-sectional surface (line 225) in Figure 4. Can you add arrows or texts on the images to point those out? That would be more indicative.

Response>:

Thank you for the suggestion. The Figure 4 has been replaced in the revised version.